# Short-Term Microplastic Exposure Impairs Cognition in Hermit Crabs

**DOI:** 10.3390/ani13061055

**Published:** 2023-03-14

**Authors:** Andrew Crump, Catherine Aiken, Eoghan M. Cunningham, Gareth Arnott

**Affiliations:** 1Institute for Global Food Security, School of Biological Sciences, Queen’s University Belfast, Belfast BT7 1NN, UK; 2Animal Welfare Science and Ethics, Pathobiology and Population Science, Royal Veterinary College, Hatfield AL9 7TA, UK; 3Queen’s University Marine Laboratory, School of Biological Sciences, Queen’s University Belfast, Portaferry BT22 1PF, UK

**Keywords:** animal cognition, hermit crab, microplastic pollution, shell selection

## Abstract

**Simple Summary:**

Little is known about how microplastics impact animal cognition—the way animals gather and process information. We investigated whether microplastic exposure impaired hermit crab shell selection, which relies on cognitive assessments and decision-making. After short-term microplastic exposure in a laboratory, hermit crabs were worse at choosing a shell. Microplastics, therefore, disrupted cognition. If wild hermit crabs are also susceptible to this effect, microplastic pollution may hinder shell selection, a crucial survival behaviour.

**Abstract:**

We tested whether acute microplastic exposure impacts information gathering and processing (cognition) in hermit crabs (*Pagurus bernhardus*). For five days, we kept 51 hermit crabs in tanks containing either polyethylene microspheres (*n* = 27) or no plastic (*n* = 24). We then transferred individuals into an intermediate-quality shell and presented them with two vials containing either a better or worse shell. Because touching both shell vials required an equivalent behavioural response, this design controlled for general activity. Plastic-exposed hermit crabs were less likely and slower than controls to touch the better shell vial, instead preferring the worse shell vial. Microplastics, therefore, impaired assessments and decision-making, providing direct evidence of acute microplastic exposure disrupting hermit crab cognition.

## 1. Introduction

Global plastic production has grown exponentially since the 1950s [1] and now exceeds 360 million tonnes per year [2]. Up to 10% of plastics ultimately enter the ocean [3,4]. Plastics < 5 mm in length or diameter are termed microplastics [5,6,7]. These are formed either as industry-made particles (primary microplastics [8]) or when larger plastics break down (secondary microplastics [9]). Among the most common ocean pollutants [10], microplastics occur from the tropics to the poles [11], and from surface waters [10] to the deep sea [12].

Microplastic pollution is a major threat to marine biodiversity [1,13]. Both direct exposure and ingestion of microplastics can negatively affect animal health and physiology [14,15,16] (but see [17]). There is also growing concern about microplastic impacts on animal behaviour. For example, microplastic exposure disrupts locomotion (zebrafish, *Danio rerio* [18]; oysters, *Crassostrea gigas* [19,20]; amphipods, *Platorchestia smithi* [21]; copepods, *Temora turbinata* [22]), feeding (amphipods, *Orchestoidea tuberculata* [23]; copepods, *Calanus helgolandicus* [24]), and shoaling behaviour (carp, *Carassius carassius* [25]).

European hermit crabs (*Pagurus bernhardus*) are an emerging model for the effects of microplastic exposure on animal behaviour [26,27]. To protect their soft abdomens from predators, these marine crustaceans live in empty gastropod shells [28,29]. Hermit crabs gather information about potential shells through visual and tactile assessments, which allow them to compare the new shell with their current shell [30,31]. Ecuadorian hermit crabs (*Coenobita compressus*) even learn to forego larger, preferred shells in favour of smaller, non-preferred shells, when the latter allow them to escape confinement through a small exit [32]. Hermit crab shell selection is, thus, a useful experimental system for studying cognition [33,34]: the acquisition, processing, retention, and use of information [35]. In hermit crabs, these processes are sensitive to environmental disturbances, including anthropogenic noise [36], ocean acidification [37], and high temperatures [38].

Microplastic exposure also disrupts crucial hermit crab behaviour (for a general review of microplastic impacts on decapod crustaceans, see [39]). Microplastics can decrease hermit crab startle durations, potentially increasing predator vulnerability [40]. In shell contests, microplastic-treated defenders needed more strikes (“raps”) from control attackers to leave their shells [27]. Microplastics may, therefore, impair hermit crabs’ cognitive assessments of attacker fighting ability.

Moreover, hermit crabs exposed to microplastics for five days were less likely and slower to upgrade a current suboptimal shell for a better alternative shell, compared to control individuals [26]. The authors speculated that microplastics impaired cognitive assessments and decision-making (see also [41]). However, this cognitive explanation was confounded, because hermit crabs had to swap their current (suboptimal) shell with a new (better) shell. Acute microplastic exposure could accordingly reduce general activity, lessening hermit crabs’ inclination to swap shells. Non-cognitive mechanisms might cause this effect, such as microplastics reducing gill function or feeding efficiency [39]. Additionally, a replication of the experiment found inconsistent results [42]. Acute microplastic exposure did not affect whether males selected the better shell, whilst plastic-exposed females chose the better shell more often than control females. These two issues (experimental confounds and failed replication) necessitate more research.

In the current study, we tested whether acute microplastic exposure affects hermit crab shell selection. Crucially, individuals began in an intermediate-quality shell and were offered a better or worse alternative shell, which both required equivalent behavioural responses. This approach controlled for general activity, so we could establish if microplastics disrupt the cognitive assessments and decision-making that underpin shell selection. A control group underwent identical shell selection trials, but without receiving acute microplastic exposure beforehand. We hypothesised that plastic-exposed hermit crabs would first touch the better shell in fewer trials than controls, and they would be slower to do so. This finding would suggest that acute microplastic exposure impaired cognitive assessments and decision-making.

## 2. Materials and Methods

### 2.1. Ethics

In the UK, the Animals (Scientific Procedures) Act 1986 does not protect crustaceans, despite the Animal Welfare (Sentience) Act 2022 recognising them as legally sentient [43,44,45]. Queen’s University Belfast’s “Animal Welfare Ethical Review Body” also does not cover crustacean research, so our study required no ethical approval. We nonetheless followed the Association for the Study of Animal Behaviour’s “Guidelines for the Use of Animals in Research”. After testing, all hermit crabs were returned to the shore without any visible signs of harm.

### 2.2. Subjects, Housing, and Treatments

From January to March 2020, we collected 51 *P. bernhardus* hermit crabs (25 males, 26 non-gravid females; weight: 0.12–1.60 g) from Ballywalter Beach, County Down, Northern Ireland, UK (54.5408° N, 5.4840° W). They were transported to Queen’s University Belfast’s animal behaviour laboratory, which was maintained at 11 °C with a 12:12 h light:dark cycle. We kept the hermit crabs in glass stock tanks (45 × 25 × 25 cm) containing approximately five litres of sand-filtered, UV-treated, constantly aerated seawater, and 50 g of bladderwrack seaweed (*Fucus vesiculosus*). The water and seaweed were entirely changed before each experimental run (i.e., once a week). To maintain consistent water quality for all subjects, we used the same stock of seawater, which was kept in sealed drums throughout the experiment. We fed the hermit crabs commercial fish food (catfish pellets) when they arrived in the lab; this food was then available ad libitum throughout the experiment.

Hermit crabs were randomly allocated to one of two treatments: either exposed to microplastics (PLAS; *n* = 27) or not (CTRL; *n* = 24). In the PLAS treatment, the glass stock tanks contained 50 g of virgin polyethylene spheres (Materialix Ltd., London, UK; diameter: 4 mm, 0.02 g [26,27]). Polyethylene is Europe’s most-produced plastic, and the most common plastic in both wild marine organisms [46] and decapod crustaceans [39]. Whilst our chosen volume of microplastics produced a relatively high concentration by weight (~5 g L^−1^), the large particle size ensured a relatively low concentration by particle number (25 particles L^−1^). Our microplastic density, thus, reflected values widely reported from coastlines [14]. No hermit crabs have died during this microplastic treatment in the present or previous studies [26,42], suggesting the volume is not fatal. In the CTRL group, the tanks contained no added microplastics. There were six stock tanks (three PLAS tanks and three CTRL tanks) to minimise any tank effects. In both treatments, hermit crabs inhabited their tank for five days before testing. Using the “pwr” package in R [47], we performed a power analysis to confirm that our sample size was large enough. For an effect size of 0.8, a power of 0.8, and a significance level of 0.05, our treatment groups of 27 and 24 were sufficient.

### 2.3. Shell Selection Testing

Following the five-day exposure period, we used a small bench-vice to crack open and remove the hermit crabs from their original shells [36]. Individuals were then dried, sexed, and weighed. Previous studies have revealed the regression lines for preferred shell weight across a range of hermit crab weights (*y* = 0.60 + 2.27*x* [28,34]), so we gave each individual a new *Littorina obtusata* shell 50% of its preferred shell weight. The hermit crabs then spent two hours in a glass crystallising dish to recover from handling and acclimate to their new shell.

After two hours, we moved hermit crabs individually into a circular glass testing tank (radius: 8 cm), filled with 7.5 cm of aerated seawater (Figure 1). Individuals began the trial 10 cm from two glass vials; one vial contained a 100% shell (better than the 50% shell), whereas the other vial contained a 25% shell (worse than the 50% shell). As a result, touching both the 100% and 25% shell vials required equal motor activity and energy expenditure. To control for potential side biases, we randomised and counterbalanced the shell location across individuals. Between each shell vial and the hermit crab starting point, there was a stained-glass screen: one black and one white. These two screens were intended as conditioned stimuli in subsequent associative learning trials but, due to time constraints imposed by the COVID-19 pandemic, additional trials were not completed.

The testing tank was inside an illuminated observation chamber, with one-way mirrors allowing us to observe the hermit crabs’ behaviour. For the categorical response variable, we recorded which shell vial the individual touched first (100% or 25%). Our continuous response variables were latency to first touch either shell vial, latency to first touch the 100% shell vial, time spent within 5 cm of both the 100% and 25% shell vials (exploration duration), and rapping duration at both the 100% and 25% shell vials. After 15 min, we ended the trial and returned the hermit crab to its stock tank. The complete dataset is provided in the Appendix A).

### 2.4. Statistics

We analysed the data in R (R Core Team, R Foundation for Statistical Computing, Vienna, Austria; version 3.6.2). To test whether hermit crabs typically first touched the 100% or 25% shell vial, we ran binary logistic regression models using the “glm” function (package: “lme4” [48]; explanatory variables: plastic treatment, sex, and their interaction). We analysed latency and duration data using two-factor ANOVAs (explanatory variables: plastic treatment, sex, and their interaction). We log-transformed the data to better meet this test’s normality assumption, and assigned a ceiling latency of 15 min if the hermit crab did not touch the shell vial. For exploration and rapping duration at each shell vial, we excluded individuals that did not touch the shell vial. We consider *p* < 0.05 significant and present the data as median, followed by interquartile range.

## 3. Results

Forty-nine hermit crabs touched at least one shell vial (100% and/or 25%). Comparing only these individuals, PLAS individuals first touched the 100% shell vial in fewer trials than CTRL individuals (*z* = 2.28, *p* = 0.02; Table 1). We found no sex difference (*z* = 0.11, *p* = 0.91) nor treatment × sex interaction (*z* = −1.21, *p* = 0.23).

Compared to CTRL hermit crabs, individuals in the PLAS treatment took longer to touch the 100% shell vial (*F*_1,45_ = 15.71, *p* < 0.001; PLAS: 301.0, 54.9–750.0 s; CTRL: 19.3, 11.8–59.1 s). There was no sex difference (*F*_1,45_ = 0.05, *p* = 0.83), but there was a treatment × sex interaction (*F*_1,45_ = 5.68, *p* = 0.02; Figure 2). The effect—longer latencies in the PLAS treatment than in the CTRL group—was more pronounced in males than females.

Additionally, PLAS hermit crabs were slower to touch a shell vial (either 100% or 25%) than CTRL individuals (*F*_1,45_ = 5.43, *p* = 0.02; PLAS: 67.6, 12.4–240.7 s; CTRL: 14.1, 7.4–59.1 s). Sex was not significantly associated with latency to first touch a shell vial (*F*_1,45_ = 1.62, *p* = 0.21). However, we recorded a significant treatment × sex interaction: males again displayed a larger treatment difference than females (*F*_1,45_ = 5.04, *p* = 0.03; Figure 3).

We also observed treatment differences in exploration duration. Compared to CTRL individuals, PLAS hermit crabs spent less time exploring the 100% shell vial (*F*_1,39_ = 10.82, *p* < 0.005; PLAS: 49.1, 17.8–227.6 s; CTRL: 299.6, 242.0–361.0 s; Figure 4). Exploration duration did not differ between males and females (*F*_1,39_ = 0.06, *p* = 0.80), and there was no treatment × sex interaction (*F*_1,39_ = 0.31, *p* = 0.58). For duration within 5 cm of the 25% shell vial, neither treatment (*F*_1,45_ = 3.84, *p* = 0.06), nor sex (*F*_1,45_ = 1.71, *p* = 0.20), nor their interaction (*F*_1,45_ = 0.37, *p* = 0.54) were significant.

Moreover, PLAS hermit crabs had shorter rapping durations at the 100% shell vial than CTRL hermit crabs (*F*_1,39_ = 15.53, *p* < 0.001; PLAS: 0.0, 0.0–1.3 s; CTRL: 13.2, 0.3–94.8 s; Figure 5a). We found no sex differences in rapping duration at the 100% shell vial (*F*_1,39_ = 0.38, *p* = 0.54) and no treatment × sex interaction (*F*_1,39_ = 1.00, *p* = 0.33). At the 25% shell vial, CTRL hermit crabs also had longer rapping durations than PLAS hermit crabs (*F*_1,45_ = 10.53, *p* < 0.005; PLAS: 0.0, 0.0–0.0 s; CTRL: 1.0, 0.0–23.3 s; Figure 5b), and female rapping durations were shorter than males’ (*F*_1,45_ = 4.30, *p* = 0.04; female: 0.0, 0.0–0.2 s; male: 1.0, 0.0–20.0 s). There was no treatment × sex interaction in the 25% shell vial rapping duration (*F*_1,45_ = 3.88, *p* = 0.05).

## 4. Discussion

Given the choice between two shell vials—one containing a better shell (100%) and the other containing a worse shell (25%) than their current shell (50%)—hermit crabs that underwent acute microplastic exposure were less likely to first touch the 100% shell vial than control hermit crabs. Crump et al. (2020) [26] also found that, compared to control individuals, plastic-treated hermit crabs were less likely and slower to swap their current 50% shell for an alternative 100% shell. By contrast, in another study on this system, acute microplastic exposure did not affect shell selection in male hermit crabs and apparently increased the likelihood of females choosing the better shell [42]. These earlier studies did not control for general activity, because the hermit crabs decided between a current shell (i.e., no motor response) and swapping for a new shell (i.e., a motor response). The present experiment eliminated this confound by comparing responses to two new shells. Both new shells were equidistant from the starting point, so touching either required equivalent motor responses. This ruled out a plastic-induced reduction in general activity, and demonstrated that acute microplastic exposure disrupted some aspect of cognition (assessment and/or decision-making). However, we do not yet know which cognitive process was affected.

Not only were PLAS hermit crabs less likely to touch the 100% shell vial than CTRL individuals, but they actually preferred the 25% shell vial. Of hermit crabs that touched a shell vial, 17/25 in the PLAS treatment touched the 25% shell vial first, compared to only 8/24 in the CTRL group. Crump et al. (2020) [26] reported a similar preference: 19/29 PLAS hermit crabs did not exchange their 50% shell for a 100% shell, versus only 10/35 CTRL hermit crabs. Unlike these two studies, McDaid et al. (2023) [42] found that only 14/40 PLAS hermit crabs failed to swap for the better shell, compared to 22/40 CTRL hermit crabs. This latter study’s control results—most CTRL subjects failing to select the 100% shell—do not align with previous studies, which established the preferred shell weight for a given weight of hermit crab [28,34]. This preference is not arbitrary: lower-quality shells reduce growth, reproduction, and survival rates [29]. One possible explanation for PLAS individuals’ unexpected preference in the present study is that adaptive shell sizes differ for hermit crabs exposed to microplastics. Perhaps plastic exposure reduces hermit crabs’ body condition (e.g., [49]), so they can only defend a smaller shell. However, we consider it unlikely that PLAS hermit crabs decided they could not defend anything better than the 25% shell, since they already inhabited a better (50%) shell and faced no competition during shell selection trials. More probable, in our view, is that plastic disrupts the adaptive cognitive processes involved in shell selection. Additionally, or alternatively, acute microplastic plastic exposure may increase boldness (see [50,51]). This could potentially cause the hermit crabs to select lower-quality (i.e., less safe) shells.

We also observed a significant plastic treatment × sex effect on shell selection behaviour. Compared to females, males displayed larger treatment differences in latency to touch the 100% shell vial. Males, therefore, seemed more susceptible to acute microplastic exposure than females. In their similar hermit crab study, McDaid et al. (2023) [42] did not observe this plastic treatment × sex interaction. A similar sex effect has nonetheless been observed in freshwater crustaceans (*Daphnia magna*) [52]. Microplastic exposure increased expression of the detoxification-related gene glutathione S-transferase in males, compared to females, suggesting that males are more sensitive. The mechanism underpinning our sex effect is unknown, but males facing danger (e.g., predators) can be bolder than females [53,54]. Perhaps acute microplastic exposure induced a similar effect, resulting in males taking longer than females to touch a 100% (i.e., safer) shell vial. To test this speculation, future researchers could replicate our experiment on hermit crabs out of their shells, which would be more vulnerable. Males would still be expected to take longer to touch the shell vials than females.

What was the mechanism underpinning our observed effects? We did not observe any ingestion of the individual particles, so we consider microplastic leachate the likely cause. Microplastic leachate impacts marine animals in various ways, from reduced body condition and reproduction [55] to impaired immunity [39] and altered behaviour [56]. Chemical leachate can also cause sensory disruption, potentially reducing PLAS hermit crabs’ ability to assess shell quality. For example, certain chemical signatures released by microplastics (e.g., dimethyl sulphide) are associated with natural trophic interactions in the marine environment, so their release can disrupt olfaction [57,58]. These chemical signatures attract hermit crabs and increase their activity levels [59]. Exposure could also impair hermit crabs’ vision [60]. Our microplastics floated on the tanks’ surface, so vision impairment would not be caused through direct contact with the eyes, but more likely through chemical leachate. To our knowledge, no studies have directly investigated the effects of microplastics or their chemical leachates on marine animal vision. However, toxic mechanisms of microplastics (e.g., disrupted amino acid metabolism) cause neurotoxicity in marine animals [61], which could cause visual nerve damage in hermit crabs. We recommend repeating our experiment with only chemical leachate, rather than microplastic particles, which would establish whether they underpin our observed effects.

This study had various limitations, which limit how far we can generalise our results. The five-day microplastic exposure period was very short compared to exposure in the natural environment. With longer exposure, hermit crabs may adapt to the presence of microplastics. Alternatively, longer exposure may increase adverse impacts on shell selection. We used an acute exposure period to maintain consistency with previous studies [26,42], thereby enabling us to compare results. We nonetheless encourage future researchers to vary the exposure period and explore potential impacts on shell selection behaviour. Another limitation was that we collected hermit crabs during winter, but aspects of the laboratory environment approximated spring (relatively long daylight hours and high temperatures). This may have impacted the animals’ physiology. However, we chose these laboratory conditions to remain consistent with previous studies on this system [26,42]. The hermit crabs also had five days to acclimate to the new environment before undergoing shell selection trials. Nonetheless, crustacean responses to odour cues exhibit sex-specific differences throughout the year (e.g., in shore crabs, *Carcinus maenas* [62]). Our observed sex effects may not, therefore, generalize to other seasons or laboratory conditions. In addition, while we attempted to ensure stable and good water quality throughout the experiments, changes in pH could have affected the hermit crabs. Reduced seawater pH can impair hermit crab shell assessments [37], so we suggest a future study investigating possible synergistic effects between both stressors.

Finally, whilst our results point to microplastics threatening wild hermit crabs, this interpretation warrants caution. Except knowing that all originated from the same beach, we have no data on individuals’ microplastic exposure before the experiment—only what they received in the treatment phase. Laboratories also differ from coastlines [14,63]. Variables such as microplastic levels, water level, temperature, and pH remained stable across our experiment, rather than fluctuating throughout the day. Daily and seasonal fluctuations in pH, for instance, can impact behavioural responses in marine organisms [64,65,66], including crustaceans [37] and involving stress-induced metabolites [67]. Such discrepancies may produce different effects in nature, so we recommend studies investigating whether our findings apply to wild hermit crab populations. For example, researchers could weigh the shells of hermit crabs from beaches with different microplastic concentrations. We hypothesise a link between higher microplastic concentrations and smaller, less optimal shells. Future studies might also address the fitness impact of this effect: do beaches with more microplastic pollution have fewer, smaller, less fecund hermit crabs? Alternatively, could there be adaptive advantages to a relatively small shell in areas with high microplastic concentrations (e.g., increased growth, fecundity, and fighting ability)?

## 5. Conclusions

Acute microplastic exposure disrupted shell selection behaviour in hermit crabs. Plastic-treated hermit crabs were less likely and slower to touch an optimal shell vial than control individuals. Indeed, acute microplastic exposure caused most hermit crabs to touch the suboptimal shell vial first. Because touching each shell vial involved an equivalent motor response, these results cannot be explained in terms of general activity. We, therefore, demonstrated that microplastic exposure negatively affected cognitive assessments and decision-making, causing maladaptive shell selection.

## Figures and Tables

**Figure 1 animals-13-01055-f001:**
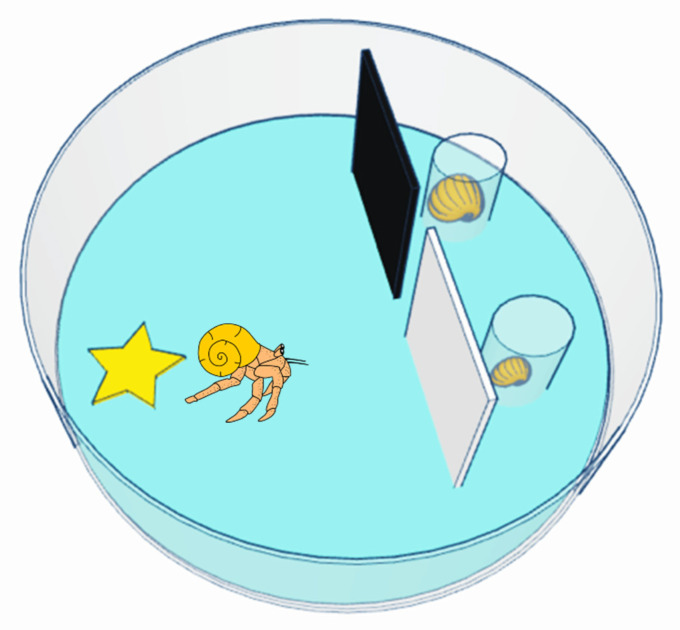
Testing tank. Hermit crabs began each trial in an intermediate-quality shell. Glass vials containing two alternative shells were located behind black/white screens. Previous studies have found that the larger shell is preferred to the intermediate-quality shell, whilst the smaller shell represents a downgrade.

**Figure 2 animals-13-01055-f002:**
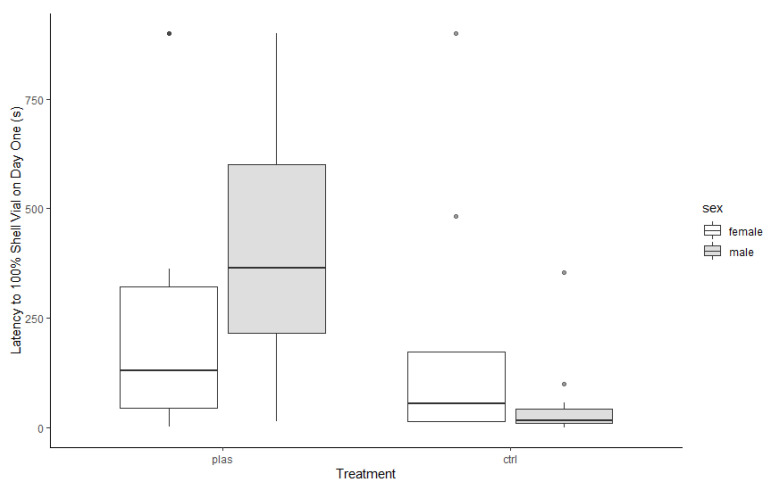
Latency for hermit crabs to touch a vial containing a better (100%) shell than their current (50%) shell. PLAS: acute microplastic exposure treatment; CTRL: control.

**Figure 3 animals-13-01055-f003:**
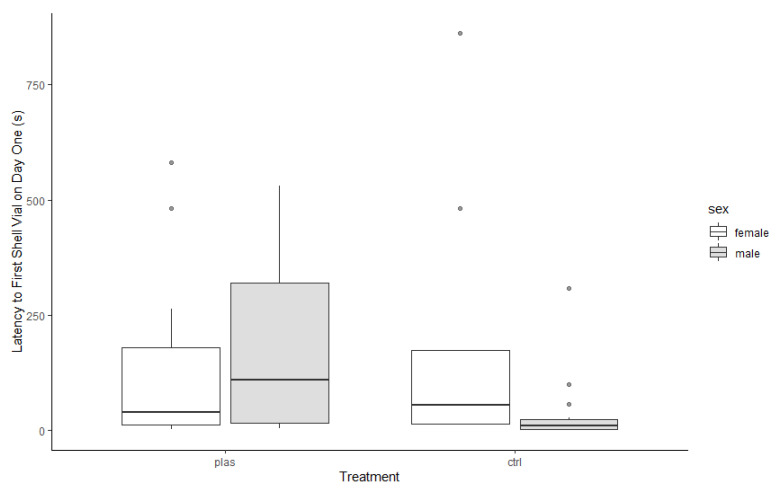
Latency for hermit crabs to touch a shell vial (either a better 100% shell or a worse 25% shell, compared to their current 50% shell). PLAS: acute microplastic exposure treatment; CTRL: control.

**Figure 4 animals-13-01055-f004:**
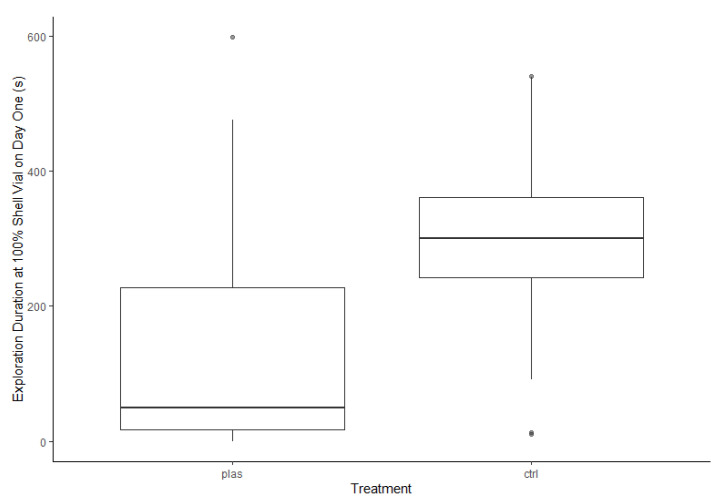
Exploration duration within 5 cm of the better (100%) shell vial. PLAS: acute microplastic exposure treatment; CTRL: control.

**Figure 5 animals-13-01055-f005:**
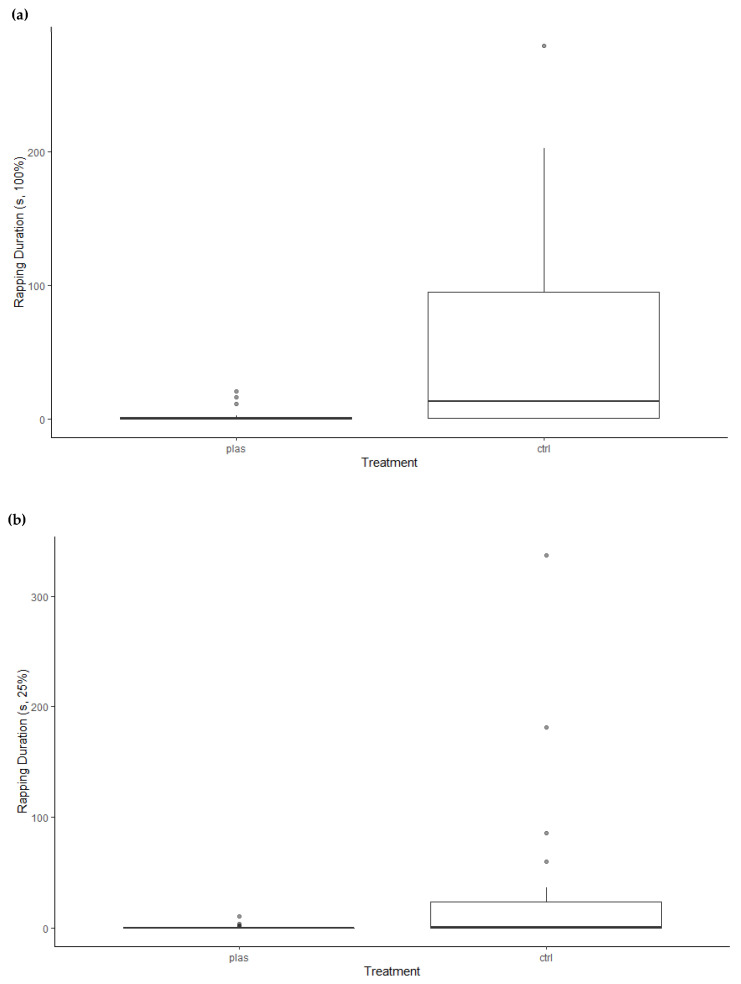
Rapping duration at the (**a**) better (100%) shell vial and (**b**) worse (25%) shell vial. PLAS: acute microplastic exposure treatment; CTRL: control.

**Table 1 animals-13-01055-t001:** Number of hermit crabs that first touched the vial containing a better (100%) or worse (25%) shell than their current (50%) shell. PLAS: acute microplastic exposure treatment; CTRL: control.

Treatment	Touched 100% First	Touched 25% First
PLAS	8	17
CTRL	16	8

## Data Availability

The data are available in the Appendix A.

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
