# Peer review of "Short-Term Microplastic Exposure Impairs Cognition in Hermit Crabs"

_animals, 2023, doi:10.3390/ani13061055_

Round 1

Reviewer 1 Report

This manuscript evaluates the impact of plastic exposure up hermit crabs. The work involved keeping the hermit crabs in plastic containing water using polyethylene pellets for 5 days and then test these hermit crabs upon their behaviour. The behaviour chosen was for the hermit crabs to choose between a large (healthy) shell and a small low-quality shell. The crabs in control did choose the large shell but when exposed to plastic the response changed to a majority picking the smaller shell.

The manuscript is quite well written and presented and should be publishable. There are though a number of questions I have mainly concerning the methods that require clarification. My main concern being the lack of replicates exemplified in the slightly disturbing statement that replication failed (line 66). I am confident the authors will be able to clarify this statement as the data presented are clear and significant. 

More specific comments to address

-       If I do understand it correctly the entire dataset consists of 51 behavioural assays with 24 and 27 crabs in the two groups as the experiments were not successfully repeated (line 66 states the repeats failed- why? Please explain)- to me this is looking on the low side and the authors really need to explain why no repeats were undertaken respectively what went wrong in the repeat.

-       It is also not entirely clear how water quality was monitored and kept at good levels. Feeding as libitum means that there were no regular defined intervals? This could influence appetite/fitness and impact the likelihood of crabs moving around in the bioassay tank. Was water salinity, ammonia and also pH controlled? A significant number of studies have shown that pH has a significant impact on hermit crab behaviour ie feeding/predator recognition so it is likely to impact the shell selection study as well. 

-       The amount of PE pellets used to make PE odour water seems excessive- 50g is a large amount given the methods (line 89) state the animals were kept in 5l tanks. Please explain as the high concentration of PE suggests toxic impacts on hermit crab fitness rather than olfactory disruption to be the basis of the observed results.

-       The discussion on sex specificity of hermit responses to PE highlights a general problem in the methods as hermit crabs used were collected in winter (Jan-March- why actually over such a long period when only 51 were used?) conditions but kept at long day and mid spring temperature conditions which will have impacted upon the physiology of the animals. It is long known that sex specific differences to odour cues exist over the season (see Hayden et al (2007) demonstrating this in shore crabs. The discussion needs to be broadened a bit and take factors impacting sex differences more into account. 

-       Equally there are other studies showing effects of microplastic and microplastic derived chemicals such as plasticisers or slipping agents upon marine life (see papers by Savoca 2016,2017 Pfaller 2020) also including  hermit crab behaviour (Greenshield et al 2021)- it would be desirable to broaden the discussion a bit and include literature on olfactory disruption in the discussion.

-       Line 207 onwards discussed the reversal of the crabs response with PE odour exposure which is generally ok but does not explain why the hermit crabs suddenly prefer the poorer quality shells. A disruption of olfactory or visual sensory cues should rather result in a 50/50 random choice as hermit crabs fail to detect the bigger, high quality shells but not in a reversal. I would at this stage add the Briffa article (48, line 223) they cite later to argue for a bold/shy change that would explain this best here too.   

-       Line 241 in the list of variable please add pH as also following daily as well as seasonal cycles and has been shown to impact behavioural responses in many marine organisms (Clements & Hunt 2015, see review by Porteus et al 2021, Roggatz et al 2022); this including a range crustaceans ( see papers by Briffa,etc) and also includes stress induced metabolites (Feugere et al 2021) all of which may well be of relevance in explaining the observed results .

Author Response

We thank all the Reviewers and the Editor for their thoughtful and incisive feedback on our manuscript. Having now completed the revisions, we believe their comments have immeasurably improved the manuscript. We provide detailed responses to the Reviewer's comments (reprinted in bold) below.

This manuscript evaluates the impact of plastic exposure up hermit crabs. The work involved keeping the hermit crabs in plastic containing water using polyethylene pellets for 5 days and then test these hermit crabs upon their behaviour. The behaviour chosen was for the hermit crabs to choose between a large (healthy) shell and a small low-quality shell. The crabs in control did choose the large shell but when exposed to plastic the response changed to a majority picking the smaller shell.

The manuscript is quite well written and presented and should be publishable. There are though a number of questions I have mainly concerning the methods that require clarification. My main concern being the lack of replicates exemplified in the slightly disturbing statement that replication failed (line 66). I am confident the authors will be able to clarify this statement as the data presented are clear and significant. 

We thank the Reviewer for their kind words about our study. We also thank them for highlighting this "failed replication" issue, which we have addressed fully below. Briefly, we have now clarified in the manuscript that “failed replication” referred to a previous study (already published; L69-72, 219-222, 236-238). As the reviewer points out, the results of the present study were clear and significant.

More specific comments to address

-       If I do understand it correctly the entire dataset consists of 51 behavioural assays with 24 and 27 crabs in the two groups as the experiments were not successfully repeated (line 66 states the repeats failed- why? Please explain)- to me this is looking on the low side and the authors really need to explain why no repeats were undertaken respectively what went wrong in the repeat.

Thank you for raising these issues. As above, the “failed replication” referred to a previously-published study (McDaid et al. 2023; described at L69-72), rather than the present experiment. Unlike the present study and another that we have carried out on this system (Crump et al. 2020), McDaid et al. found inconsistent results. Acute microplastic exposure did not affect whether males selected the better shell, whilst plastic-exposed females chose the better shell more often than control females. We have now edited the Discussion to clarify our meaning (L219-222, 236-238).

Whilst we appreciate the reviewer’s point about sample size, we confirmed that our sample size was sufficient using a power analysis. Using the R package “pwr”, we determined that our sample sizes of 24 and 27 were high enough to produce a significant result at P = 0.05 with a power value of 0.8 and an effect size of 0.8. We have now added this to the Methods (L120-122).

-       It is also not entirely clear how water quality was monitored and kept at good levels. Feeding as libitum means that there were no regular defined intervals? This could influence appetite/fitness and impact the likelihood of crabs moving around in the bioassay tank. Was water salinity, ammonia and also pH controlled? A significant number of studies have shown that pH has a significant impact on hermit crab behaviour ie feeding/predator recognition so it is likely to impact the shell selection study as well. 

While water quality was not directly monitored during the 5-day experiments, best efforts were made to ensure it was kept at good levels. All seawater used during the experiments was collected on the same day from the Queen’s Marine Laboratory using the UV-treated and sand-filtered seawater system. This stock supply of seawater was stored in drums so salinity could not be altered by evaporation during the experiments. Full water changes were carried out on the experimental and animal stock tanks at the end of each experimental run (5 days). Each behavioural observation was carried out in crystalising dishes containing fresh seawater from the same stock. We mentioned that food was available ad libitum; however, feeding did occur on a consistent schedule before the 5-day exposure period. We have now clarified these points in the Methods (L100-106).

We absolutely agree that pH can affect the behaviour of hermit crabs, so we have added this as a potential limitation in the Discussion (L298-301, 308-310). However, it is likely that our regular and thorough tank maintenance prevented pH from causing the effect found in our study. In addition, multiple tanks for each treatment were also used to avoid any potential confounds as a result of “tank effects” such as reduced water quality.

-       The amount of PE pellets used to make PE odour water seems excessive- 50g is a large amount given the methods (line 89) state the animals were kept in 5l tanks. Please explain as the high concentration of PE suggests toxic impacts on hermit crab fitness rather than olfactory disruption to be the basis of the observed results.

Thank you for raising this issue, which we have now clarified in the Methods. Whilst the reviewer is correct that our chosen volume of microplastics produced a relatively high concentration by weight (~5 g l−1), the large particle size (2mm diameter) ensured a relatively low concentration by particle number (25 particles l−1; L112-114). Indeed, by particle number, our microplastic density reflected values widely reported from coastlines (Cunningham & Sigwart 2019; L114-115). Moreover, no hermit crabs have died during this microplastic treatment in the present or previous studies (Crump et al. 2020, McDaid et al. 2023), suggesting that the volume is not fatal (L115-117).

-       The discussion on sex specificity of hermit responses to PE highlights a general problem in the methods as hermit crabs used were collected in winter (Jan-March- why actually over such a long period when only 51 were used?) conditions but kept at long day and mid spring temperature conditions which will have impacted upon the physiology of the animals. It is long known that sex specific differences to odour cues exist over the season (see Hayden et al (2007) demonstrating this in shore crabs. The discussion needs to be broadened a bit and take factors impacting sex differences more into account.

We agree with the Reviewer that this issue should be addressed in the Discussion, so we have discussed it at length in the new limitations paragraph (L290-298). Briefly, we chose these laboratory conditions to remain consistent with our previous studies on this system (Crump et al. 2020, McDaid et al. 2023), thereby allowing us to reasonably compare the results. Hermit crabs also had five days to acclimate to the new environment before undergoing shell selection trials. However, as the Reviewer highlights, our findings may not generalise to other times of the year or laboratory conditions.

The Reviewer also asks why only 51 hermit crabs were tested. As with the aborted associative learning trials, this was due to time constraints imposed by the COVID-19 pandemic, which closed our labs for all non-essential research. As above, we nonetheless carried out a power analysis, which confirmed that our sample size was sufficient. We have now added this information to the manuscript (L120-122).

-       Equally there are other studies showing effects of microplastic and microplastic derived chemicals such as plasticisers or slipping agents upon marine life (see papers by Savoca 2016,2017 Pfaller 2020) also including hermit crab behaviour (Greenshield et al 2021)- it would be desirable to broaden the discussion a bit and include literature on olfactory disruption in the discussion.

We agree, and have now added a paragraph to the Discussion, which considers the potential mechanism underpinning our observed effects (L266-282). As the Reviewer helpfully recommends, this considers the effects of microplastic-associated chemicals on marine life and hermit crabs in detail. We suggest chemical leachate disrupting olfaction as a potential mechanism for our observed effects.

-       Line 207 onwards discussed the reversal of the crabs response with PE odour exposure which is generally ok but does not explain why the hermit crabs suddenly prefer the poorer quality shells. A disruption of olfactory or visual sensory cues should rather result in a 50/50 random choice as hermit crabs fail to detect the bigger, high quality shells but not in a reversal. I would at this stage add the Briffa article (48, line 223) they cite later to argue for a bold/shy change that would explain this best here too.

Thank you for this suggestion, which we have now included in the manuscript (L248-251).

-       Line 241 in the list of variable please add pH as also following daily as well as seasonal cycles and has been shown to impact behavioural responses in many marine organisms (Clements & Hunt 2015, see review by Porteus et al 2021, Roggatz et al 2022); this including a range crustaceans ( see papers by Briffa,etc) and also includes stress induced metabolites (Feugere et al 2021) all of which may well be of relevance in explaining the observed results.

Thank you for this excellent point and the references. We have now added pH to this list (L306) and explained its importance in the Discussion (L308-310).

Reviewer 2 Report

The manuscript describes the effect of microplastic exposure on hermit crabs' shell choice. The experiments are based on shell size recognition and shell selection through vision. It is a good manuscript, interesting, and well-written.

I have just a main comment. The manuscript requires that the authors describe the action mechanism and target organ/systems of microplastic toxicity. The study predicts and discusses microplastic's effect on hermit crabs' cognition, but the hermit crabs were forced to recognize the shell´s size through vision. However, neither the introduction nor the discussion describes the potential impact of microplastics on animals' eyes or sight. Therefore, I suggest the authors include a brief description of the microplastic effects on crustaceans' tissues, sensory systems, and physiology (specifically on vision).  Could the shell choice be biased by vision impairment?

L-41. How can microplastics affect visual and tactile assessments of shells?

L.73. The authors haven't yet defined which condition the authors used as a control (the readers have not read the experimental design yet). Therefore, the hypothesis should be more specific and compare plastic-exposed vs. not exposed. Additionally, the hypothesis is presented more as a prediction than a hypothesis.

L-82. To affirm it, the authors require a citation stating that microplastics do not cause long-term effects.

L-220. Could the microplastic affect vision? I consider it very important that the authors describe the main toxic mechanism of microplastics.

L-243.  In addition to the future investigations about potential studies that could be conducted on microplastics and hermit crabs, I consider it important to assess if occupying a relatively small shell could be adaptive (likely increasing growth, fecundity, and fighting ability).

Author Response

We thank all the Reviewers and the Editor for their thoughtful and incisive feedback on our manuscript. Having now completed the revisions, we believe their comments have immeasurably improved the manuscript. We provide detailed responses to the Reviewer's comments (reprinted in bold) below.

The manuscript describes the effect of microplastic exposure on hermit crabs' shell choice. The experiments are based on shell size recognition and shell selection through vision. It is a good manuscript, interesting, and well-written.

I have just a main comment. The manuscript requires that the authors describe the action mechanism and target organ/systems of microplastic toxicity. The study predicts and discusses microplastic's effect on hermit crabs' cognition, but the hermit crabs were forced to recognize the shell´s size through vision. However, neither the introduction nor the discussion describes the potential impact of microplastics on animals' eyes or sight. Therefore, I suggest the authors include a brief description of the microplastic effects on crustaceans' tissues, sensory systems, and physiology (specifically on vision).  Could the shell choice be biased by vision impairment?

Thank you for this excellent point, which Reviewer 1 also raised. As above, we have now added a paragraph to the Discussion, where we consider the potential mechanism and describe microplastic toxicity in marine animals (L266-282). We have also proposed vision impairment as a possible mechanism for our observed effects (L274-280).

L-41. How can microplastics affect visual and tactile assessments of shells?

We appreciate this excellent question. Microplastics can cause visual damage in marine animals through the direct contact of particles with the eyes (see De Marco et al. 2022). However, the effects of microplastic exposure on vision (possibly because of reduced body condition and impaired physiology) remain unknown. We have now discussed this in the manuscript (L274-280).

L.73. The authors haven't yet defined which condition the authors used as a control (the readers have not read the experimental design yet). Therefore, the hypothesis should be more specific and compare plastic-exposed vs. not exposed. Additionally, the hypothesis is presented more as a prediction than a hypothesis.

Thank you for highlighting this issue. We have now added a sentence to explain the control condition (L79-80).

L-82. To affirm it, the authors require a citation stating that microplastics do not cause long-term effects.

To our knowledge, no studies have confirmed that acute microplastic exposure does not cause any adverse long-term effects in hermit crabs (e.g., D’Costa 2022 concluded that more research was needed in this area). We have, therefore, edited the ethics statement to clarify that hermit crabs were returned to the shore “without any visible signs of harm” (L93).

L-220. Could the microplastic affect vision? I consider it very important that the authors describe the main toxic mechanism of microplastics.

As above, we have now described the toxic mechanisms of microplastics found in marine animals and how they could relate to visual impairment in our Discussion (L266-282).

L-243.  In addition to the future investigations about potential studies that could be conducted on microplastics and hermit crabs, I consider it important to assess if occupying a relatively small shell could be adaptive (likely increasing growth, fecundity, and fighting ability).

Thank you. We agree, and have now added a sentence acknowledging this as an important question for future studies to address (L316-318).

Reviewer 3 Report

as attatched

Author Response

We thank all the Reviewers and the Editor for their thoughtful and incisive feedback on our manuscript. Having now completed the revisions, we believe their comments have immeasurably improved the manuscript. We provide detailed responses to the Reviewer's comments (reprinted in bold) below.

Title: Change the title to make it more catchy

Thank you for this suggestion. We have now edited the title to improve clarity (L2).

Simple summary: simple summary is written very causal and it needs through rephrasing

We appreciate the reviewer’s feedback, and have edited the Simple Summary to provide more information about the study’s aim and experimental design (L10-16). In general, though, we believe that Simple Summaries should be written in a way that’s accessible to a lay audience. We also think ours satisfies the journal’s guidelines, containing “a clear statement of the problem addressed [L10-11], the aims and objectives [L11-12], pertinent results [L13-14], conclusions from the study [L14] and how they will be valuable to society [L15-16].”

Abstract: The five days exposure to Evaluate the impact of microplastic is very short and statistically it may also not be correct.

The Reviewer makes an excellent point regarding the short duration of microplastic exposure. We selected this time period to maintain consistency with previous studies (Crump et al. 2020, McDaid et al. 2023, Nanninga et al. 2020, 2021, Villegas et al. 2022), which enabled us to compare the Results. Nonetheless, we have recognised the acute exposure period as a limitation in the Discussion, and considered its implications for our results (L284-290). Throughout the manuscript, we have also clarified that the exposure period was “acute” or “short-term”.

In abstract section a flow is missing so needs to be rephrased.

We have made some minor edits to the abstract, which hopefully improve its flow.

Introdusction

Its not having update references i.e. Sarkar et al 2021, 2022 are one of the important update. I suggest authors to put 80 % references should be beyond 2018.

We have added various recent references, including the excellent Sarker et al. papers that the reviewer suggests (L31; also: Delaeter et al. 2022, De Marco et al. 2022, Feugere et al. 2021, Greenshields et al. 2021, Huang et al. 2021, Porteus et al. 2021, Roggatz et al. 2022, Seuront 2018). However, we respectfully disagree that 80% should have been published since 2018. This may be a good rule-of-thumb in some fields, such as medicine, where numerous studies are published and papers rapidly become outdated. Neither feature applies to animal behaviour and cognition research. For hermit crab studies, in particular, only a handful of relevant studies have been published since 2018 (all of which, to our knowledge, we have already cited). An arbitrary 2018 threshold would mean removing many references that provide critical background (e.g., Briffa & Dallaway 2007, Elwood et al. 1979, Elwood & Neil 1992, Lancaster 1990).

Discussion and results have a losses coherence so discussion need significant changes as per coherent between results and discussion.

Thank you. In line with suggestions from the other reviewers, we have substantially edited the Discussion to better contextualise the Results.

Round 2

Reviewer 3 Report

The changes are being made as per the comments. So now it can be processed further.